# Association of egg consumption, metabolic markers, and risk of cardiovascular diseases: A nested case-control study

Lang Pan[1], Lu Chen[1], Jun Lv[1,2,3], Yuanjie Pang[1], Yu Guo[4], Pei Pei[5], Huaidong Du[6,7], Ling Yang[6,7], Iona Y Millwood[6,7], Robin G Walters[6,7], Yiping Chen[6,7], Weiwei Gong[8], Junshi Chen[9], Canqing Yu[1,2]*, Zhengming Chen[7], Liming Li[1,2], on behalf of China Kadoorie Biobank Collaborative Group

[1]Department of Epidemiology and Biostatistics, Peking University, Beijing, China; [2]Peking University Center for Public Health and Epidemic Preparedness & Response, Beijing, China; [3]Key Laboratory of Molecular Cardiovascular Sciences (Peking University), Ministry of Education, Beijing, China; [4]Fuwai Hospital Chinese Academy of Medical Sciences, National Center for Cardiovascular Diseases, Beijing, China; [5]Chinese Academy of Medical Sciences, Beijing, China; [6]Medical Research Council Population Health Research Unit at the University of Oxford, Oxford, United Kingdom; [7]Clinical Trial Service Unit & Epidemiological Studies Unit (CTSU), Nuffield Department of Population Health, University of Oxford, Oxford, United Kingdom; [8]NCDs Prevention and Control Department, Zhejiang CDC, Hangzhou, China; [9]China National Center for Food Safety Risk Assessment, Beijing, China

*For correspondence:
yucanqing@pku.edu.cn

Competing interest: The authors declare that no competing interests exist.

## Abstract

**Background:** Few studies have assessed the role of individual plasma cholesterol levels in the association between egg consumption and the risk of cardiovascular diseases. This research aims to simultaneously explore the associations of self-reported egg consumption with plasma metabolic markers and these markers with the risk of cardiovascular disease (CVD).

**Methods:** Totally 4778 participants (3401 CVD cases subdivided into subtypes and 1377 controls) aged 30–79 were selected based on the China Kadoorie Biobank. Targeted nuclear magnetic resonance was used to quantify 225 metabolites in baseline plasma samples. Linear regression was conducted to assess associations between self-reported egg consumption and metabolic markers, which were further compared with associations between metabolic markers and CVD risk.

**Results:** Egg consumption was associated with 24 out of 225 markers, including positive associations for apolipoprotein A1, acetate, mean HDL diameter, and lipid profiles of very large and large HDL, and inverse associations for total cholesterol and cholesterol esters in small VLDL. Among these 24 markers, 14 were associated with CVD risk. In general, the associations of egg consumption with metabolic markers and of these markers with CVD risk showed opposite patterns.

**Conclusions:** In the Chinese population, egg consumption is associated with several metabolic markers, which may partially explain the protective effect of moderate egg consumption on CVD.

**Funding:** This work was supported by the National Natural Science Foundation of China (81973125, 81941018, 91846303, 91843302). The CKB baseline survey and the first re-survey were supported by a grant from the Kadoorie Charitable Foundation in Hong Kong. The long-term follow-up is supported by grants (2016YFC0900500, 2016YFC0900501, 2016YFC0900504, 2016YFC1303904) from the National Key R&D Program of China, National Natural Science Foundation of China

(81390540, 81390541, 81390544), and Chinese Ministry of Science and Technology (2011BAI09B01). The funders had no role in the study design, data collection, data analysis and interpretation, writing of the report, or the decision to submit the article for publication.

## Editor's evaluation

Pan et al. investigated associations of self-reported egg consumption with plasma metabolic markers and these plasma metabolic markers with the risk of cardiovascular diseases. In general, there was some impact on metabolic markers which could protect against CVD. The paper will interest scientists in the field of nutritional epidemiology.

## Introduction

Cardiovascular disease (CVD) has always been a major public health challenge and is the leading cause of death and disability worldwide, including in China (*Mortality and Death, 2016*). Ischemic heart disease (IHD), ischemic stroke (IS), and intracerebral hemorrhage (ICH) accounted for the majority of the deaths caused by CVD, which becomes more and more serious as the aggravation of population aging (*Mortality and Death, 2016*; *Wang et al., 2017*). In recent decades, with the concern about the burden of CVD, researchers not only focus on the modifiable risk factors, including smoking, drinking, physical activity, and diet, but also pay more and more attention to the internal biological mechanism (*Virani et al., 2020*; *Boehme et al., 2017*). Plasma lipids, especially low-density lipoprotein cholesterol (LDL-C) which may accumulate in the arterial wall, gradually form atherosclerotic plaques and block the corresponding artery, is generally considered to be associated with risks of IHD, IS, and ICH (*Varbo and Nordestgaard, 2014*; *Arora et al., 2019*; *Sun et al., 2019*).

Eggs are one of the richest sources of dietary cholesterol, but they also contain a wide variety of essential nutrients and bioactive compounds, such as high-quality protein, fat-soluble and B vitamins, phospholipids, and choline (*Andersen, 2015*; *Blesso, 2015*). Evidence for the association between eggs and CVD remains controversial in both observational and prospective studies based on Western populations, with positive association being reported in some studies (*Zhong et al., 2019*) and others finding no significant association (*Shin et al., 2013*; *Guo et al., 2018*; *Drouin-Chartier et al., 2020*; *Dehghan et al., 2020*; *Rong et al., 2013*). Similar controversial findings have been found in the Chinese population. An inverse association between egg consumption and CVD risk was published,[16] whereas more (>10 eggs per week) and less (<1 per week) egg intake was found to be harmful to cardiovascular health (*Xia et al., 2020*). With regret, few studies have assessed the role of individual plasma cholesterol levels in such association, making the association clearer. On the other hand, Lipoproteins can be divided into large, medium and small subclasses according to particle size and include a series of constituents including cholesterol, phospholipids, triglycerides, and apolipoprotein which is difficult for conventional approaches to quantitate. Nuclear magnetic resonance (NMR) metabolomics provides a new opportunity to explore the associations between exposure, diseases and lipoprotein and other small-molecule metabolic markers in a more detailed perspective (*Nagana Gowda and Raftery, 2019*).

The present study selected a targeted NMR-based metabolomics platform that previous large-scale studies have widely used, (*Würtz et al., 2017*) covering 14 lipoproteins and their subfractions with different densities and particle sizes, as well as other fatty acids, amino acids, and ketone-body-related metabolites. We aimed to simultaneously explore the associations of self-reported egg consumption with these metabolic markers, and of these markers with CVD risk in a nested case-control study in the China Kadoorie Biobank (CKB).

## Materials and methods
### Participants and study design

The CKB study was a prospective cohort of 512,725 participants aged 30–79 years from 5 urban and 5 rural areas across China. Participants were recruited between 2004 and 2008, whose morbidity and mortality have been followed up ever since. A laptop-based questionnaire was used to collect

detailed information, including demographic characteristics (e.g. age at recruitment, sex, education, household income, and marital status), lifestyle factors (e.g. smoking and drinking habits, food intake, and physical activities), medical history (e.g. hypertension, diabetes, and use of certain specific medications such as statins), and family history of diabetes, heart attack or stroke. Each participant also underwent a range of physical measurements operated by trained staff, including anthropometry, lung function, blood pressure, and heart rate, etc. All participants provided a 10 mL non-fasting (with time since last meal recorded) blood sample for immediate on-site random blood glucose (RBG) test and long-term storage. In addition, every 5 years after completing the baseline survey, about 5% of the participants were randomly selected to join in the re-survey. Detailed descriptions of the CKB study have been previously published (*Chen et al., 2011*).

The present study selected 4778 participants from a previous nested case-control study based on CKB (*Holmes et al., 2018*). Cases consisted of incident cases of myocardial infarction (MI, ICD-10 I21-23, n=946), IS (I63 and I69.3, n=1,217), and ICH (I61 and I69.1, n=1,238), with a censoring date of January 01, 2015. And 1377 controls were frequently matched to the combined cases by age, sex, and area if possible. All cases and controls had no history of self-reported prior doctor-diagnosed coronary heart disease (CHD), stroke, transient ischemic attack, or cancer, and were not using statin therapy at baseline. The Ethical Review Committee of the Chinese Center for Disease Control and Prevention (Beijing, China, 005/2004) and the Oxford Tropical Research Ethics Committee, University of Oxford (UK, 025–04) approved the study. Before enrolled the study, each participant signed the informed consent and agreed that the data would be used for scientific research and subsequent publication.

## Assessment of egg consumption

Using the face-to-face laptop-based food frequency questionnaire (FFQ) at baseline, participants were asked by trained investigators about their frequency of habitual egg consumption as well as other 11 food groups (rice, wheat, other staple food, meat, poultry, fish, fresh vegetables, preserved vegetables, fresh fruits, soybean products, and dairy) during the past 12 months. Possible answers were 'never/rarely, monthly, 1–3 days per week, 4–6 days per week, and daily'. The frequency was then converted into days of egg consumption per week, with each option corresponding to 0, 0.5, 2, 5, and 7 days per week, respectively. Participants were also asked about the daily amount when consuming eggs at the 2nd re-survey (2013–2014).

A separate validation study was conducted from 2015–2016 among 432 CKB participants to evaluate the reproducibility and validity of FFQ.*Qin et al., 2022* It turned out that the weighted Kappa statistic was 0.77 and 0.65 for reproducibility and relative validity of baseline eggs frequency, respectively.

## Measurement of NMR metabolisms

After centrifuging and aliquoting, baseline plasma samples from each participant were couriered from the regional laboratory via Beijing to Oxford for long-term storage in liquid nitrogen tanks. The stored plasma samples of the cases and controls were thawed and sub-aliquoted at the Wolfson laboratory, CTSU, before 100 μL aliquots being shipped on dry ice to the Brainshake Laboratory at Oulu, Finland, for high-throughput targeted NMR spectroscopy to quantify 225 absolute concentrations or derived traits (e.g. lipids ratios) of metabolic markers simultaneously (*Soininen et al., 2015*).

The samples were handled in 96-well plates containing two quality control samples (a plasma mimic and a mixture of two low-molecular-weight metabolites). The former was used to monitor the consistency of quantifications, whereas the latter was a technical reference to monitor the performance of the automated liquid handler and the spectrometer. Barcoding was preferred for sample identification, and all the liquid handling steps were done with an automated workstation. The spectral information underwent various comparisons with the spectra of the two quality control samples. Regression modeling was performed to produce the quantified molecular data for those spectral areas that passed all the quality control steps.

Samples from cases and controls were quantified randomly, with laboratory staff blinded to case or control status. The sample size of analyses of some metabolic markers involved was less than 4778 since the quality control process rejected results of these metabolic markers among some participants.

## Statistical analysis

Baseline characteristics of participants were presented as means or percentages across controls and three subtypes of CVD cases, standardized by age, sex, and study region if appropriate, using multiple linear regressions for continuous variables or logistic regressions for categorical variables.

For metabolites whose measurements below the limit of detection were imputed with the lowest measured concentration. Each metabolic marker was log-transformed and divided by its standard deviation (SD). Linear regression was used to assess the associations of egg consumption with metabolic markers, adjusted for age (years, continuous), sex, region (10 regions), education (6 categories: no formal school, primary school, middle school, high school, technical school or college, and university), household income (6 categories: <2500, 2500–4999, 5000–9999, 10,000–19,999, 20,000–34,999, and ≥35,000 Yuan per year), occupation (10 categories: agriculture and related worker, factory worker, administrator or manager, professional or technical, sales & service worker, retired, house wife or husband, self-employed, unemployed, and other or not stated), marital status (four categories: married, separated or divorced, widowed, and never married), tea-drinking habit (five categories: never or rarely, occasionally, seasonly, monthly, and weekly tea-drinker), smoking status (five categories: never or occasional, former, ≤10, ≤20, or more per day), alcohol intake (five categories: never or occasional, former, ≤25 g for men or ≤15 g for women, ≤50 g for men or ≤30 g for women, or more per day), physical activity (MET-h/d, continuous), self-rated health (poor or not), fasting time (hours, continuous) and frequency of other 11 food groups. For each biomarker, adjusted SD differences of log-transformed metabolic markers and 95% confidence intervals (CI) associated with an extra day of egg consumption per week were estimated.

Logistic regression was used to estimate odds ratios (ORs) for CVD and its three subtypes (MI, IS, and ICH) per SD higher log-transformed metabolic markers, with the same variables adjusted for as in the analysis of egg consumption and metabolic markers. ICH cases were excluded from the analysis of metabolic markers and CVD because there were no associations of metabolic markers with ICH (*Holmes et al., 2018*). ORs were then plotted against SD differences in corresponding log-transformed metabolic biomarkers per extra day of egg consumption.

In order to examine the robustness of associations between egg consumption and metabolic markers, we performed several sensitivity analyses: additionally adjusting for body mass index (BMI), presence of prevalent hypertension or diabetes, and family history of diabetes or CVD; and excluding participants with any metabolic markers below the limit of detection or rejected by quality control. In addition, participants were stratified by sex, age group (<60 y or ≥60 y), region (10 regions) and egg consumption group (never/rarely, monthly, 1–3 d/w, 4–6 d/w, or daily), and the mean values of amount on days when consuming eggs for each stratum collected in the 2nd re-survey among 23,974 participants was calculated as a proxy to estimate the amount of egg consumption (*Qin et al., 2018*). We reran the analysis using the weekly amount of egg consumption as a continuous independent variable instead of its frequency.

All p-values were two-sided, and statistical significance was defined as $p < 0.05$. To account for a large number of highly correlated metabolic markers, we calculated the false discovery rate (FDR) $p < 0.05$ based on the Benjamini-Hochberg method for associations of egg consumption with metabolic markers and of metabolic markers with risk of CVD. Statistical analyses were performed using Stata 15.0.

## Results

Age-, sex-, and region-adjusted baseline characteristics of the 4778 participants according to whether they developed CVD are shown in *Table 1*. Briefly, the mean (SD) age was 47.0 (8.2) years, 50.1% were women, and 29.0% resided in urban areas. The mean (SD) frequency of egg consumption and plasma total cholesterol concentration were 2.6 (2.3) days/week and 3.5 (0.6) mmol/L, respectively. Compared with controls, participants who subsequently developed any subtype of CVD were more likely to have poor self-rated health, to prevalent obesity, diabetes, and hypertension, and to have a family history of diabetes or CVD. Among them, MI or IS cases had a lower level of education but a higher level of household income, whereas ICH cases had higher levels of SBP and DBP.

Among the 225 metabolic markers or derived traits, 24 were associated with the frequency of egg consumption at FDR <5% (*Figure 1*). The associations for all 225 markers are shown in *Supplementary*

**Table 1.** Baseline characteristics among 4,778 participants.

| | Controls | MI cases | IS cases | ICH cases |
|---|---|---|---|---|
| N | 1,377 | 946 | 1217 | 1238 |
| Age, y | 46.87 (0.20) | 52.42 (0.24) | 42.51 (0.22) | 47.38 (0.21) |
| Female, % | 50.26 | 40.40 | 55.87 | 51.54 |
| Urban residents, % | 27.13 | 28.90 | 36.88 | 23.37 |
| Middle school or above, % | 56.97 | 53.84 | 53.70 | 56.36 |
| Income ≥35,000 Yuan/year, % | 9.98 | 12.84 | 12.89 | 10.60 |
| Manual worker, % | 69.88 | 68.39 | 69.12 | 71.50 |
| BMI, kg/m$^2$ | 23.56 (0.09) | 24.03 (0.12) | 24.27 (0.10) | 24.20 (0.10) |
| SBP, mmHg | 127.94 (0.64) | 136.96 (0.82) | 136.46 (0.72) | 149.76 (0.68) |
| DBP, mmHg | 76.97 (0.36) | 82.05 (0.47) | 82.57 (0.41) | 89.35 (0.39) |
| RBG, mmol/L | 5.63 (0.08) | 6.47 (0.10) | 6.19 (0.08) | 6.31 (0.08) |
| Ever regular smoking, % | 34.60 | 40.79 | 38.32 | 36.10 |
| Weekly drinking, % | 18.11 | 15.27 | 18.77 | 20.68 |
| Physical activity, MET-h/d | 23.20 (0.36) | 22.08 (0.46) | 22.72 (0.40) | 23.26 (0.38) |
| Eggs consumption, d/w | 2.69 (0.06) | 2.44 (0.07) | 2.62 (0.07) | 2.52 (0.06) |
| Red meat consumption, d/w | 3.26 (0.06) | 3.23 (0.07) | 3.35 (0.06) | 3.31 (0.06) |
| Fresh fruits consumption, d/w | 2.32 (0.05) | 1.91 (0.07) | 2.08 (0.06) | 2.04 (0.06) |
| ≥8 h of fasting, % | 13.65 | 15.04 | 16.69 | 14.86 |
| Poor self-rated health, % | 9.60 | 14.00 | 14.11 | 13.49 |
| Diabetes, % | 5.27 | 10.86 | 8.74 | 8.21 |
| Hypertension, % | 27.06 | 44.47 | 45.00 | 64.66 |
| Family history of diabetes, % | 7.22 | 9.02 | 9.11 | 7.98 |
| Family history of CVD, % | 23.68 | 27.55 | 26.71 | 31.04 |

Results were standardized by age, sex, and region where appropriate. Values are means (standard errors, SE) or %. Abbreviations: BMI, body mass index; SBP, systolic blood pressure; DBP, diastolic blood pressure; RBG, random blood glucose; MET, metabolic equivalent; CVD, cardiovascular disease; MI, myocardial infarction; IS, ischemic stroke; ICH, intracerebral hemorrhage.

*file 1a*. Egg consumption was positively associated with lipoprotein particle concentrations of very large and large HDL. Similarly, within very large and large HDL, there were positive associations of total lipids, total cholesterol including its subclasses (cholesterol esters and free cholesterol), or phospholipids with egg consumption. Conversely, there was an inverse association of cholesterol esters in small VLDL with egg consumption. In addition to the absolute concentrations of lipids, the percentage of total cholesterol and cholesterol esters in large HDL were positively associated with egg consumption. Besides lipids, there were positive associations of acetate and apolipoprotein A1 with egg consumption, whereas an inverse association was observed for the ratio of apolipoprotein B to apolipoprotein A1.

Among 225 metabolic markers or derived traits, 104, 123, and 55 were associated with CVD and its subtypes, MI and IS, respectively, at FDR <5% (*Supplementary file 1*). For total cholesterol within small VLDL, each SD increment in log-transformed value was associated with increased risks of CVD (OR 1.15 [95%CI 1.07–1.24]), MI (1.25 [1.14–1.38]), and IS (1.15 [1.05–1.27]), whereas the association with total cholesterol within large HDL was in the opposite direction (0.88 [0.81–0.95] for CVD and 0.83 [0.75–0.91] for MI). However, only one metabolic marker, glucose, showed a positive association with the risk of ICH. ORs for MI, IS, and ICH associated with all 225 metabolic markers were provided in *Supplementary file 1b*. Among markers associated with risk of CVD, MI, or IS, 14, 15, and 3 were simultaneously associated with egg consumption. There was a clear pattern between the associations of egg consumption with metabolic markers and of these metabolic markers with disease risk; that is,

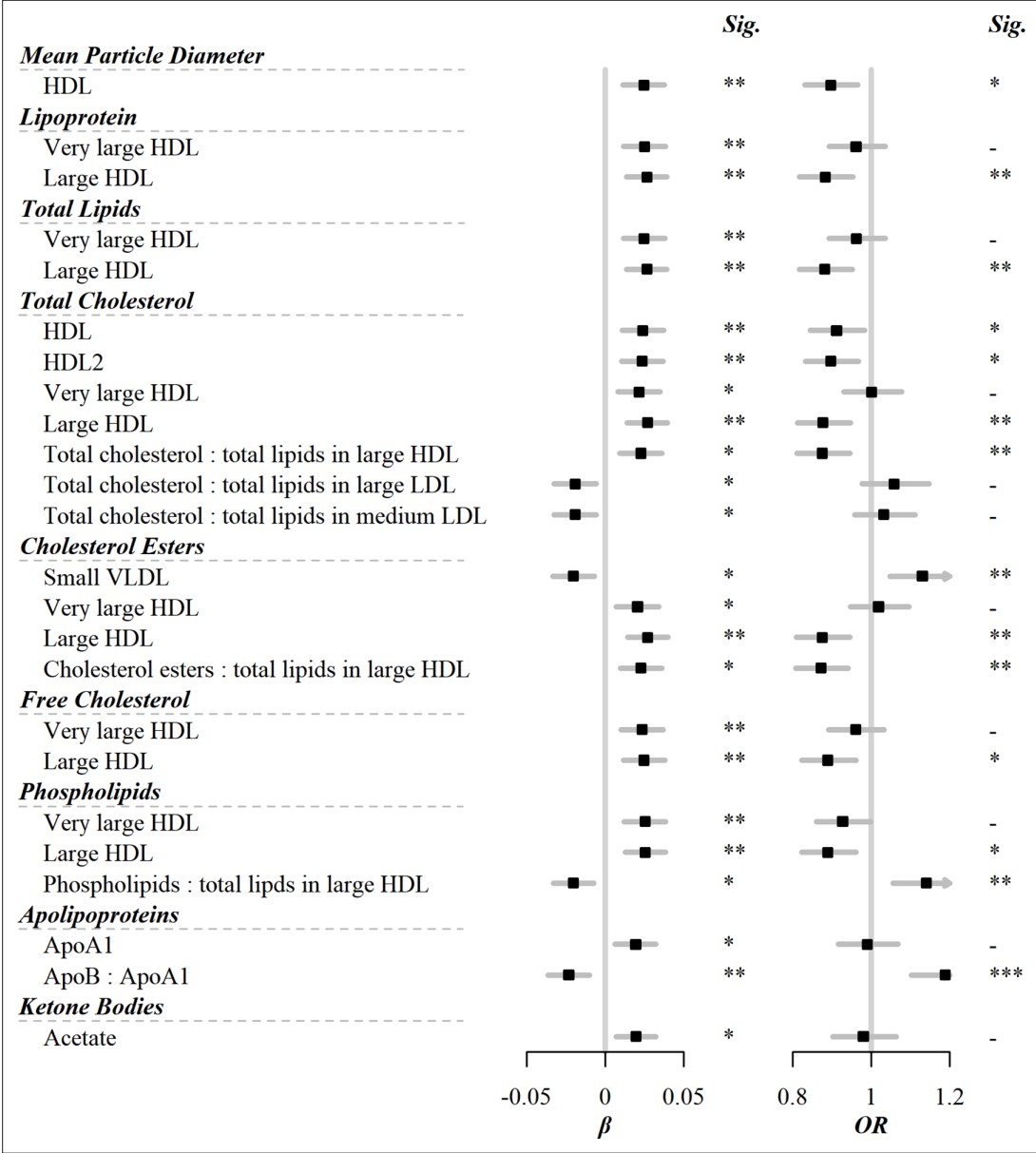

**Figure 1.** Significant associations of egg consumption and metabolic markers, and associations of these markers with risks of CVD. Models were adjusted for age, sex, region, education, household income, occupation, marital status, tea-drinking habit, smoking status, alcohol intake, physical activity, self-rated health, fasting time, and frequency of other 11 food groups. Black squares represented coefficients or ORs, while gray horizontal lines represented 95% CI. Significance (Sig.): *p<0.05, **p<0.01, ***p<0.001 (FDR-adjusted p using the Benjamini-Hochberg method). The source data can be found in .

The online version of this article includes the following source data for figure 1:

**Source data 1.** Significant associations of egg consumption and metabolic markers, and associations of these markers with risks of CVD.

metabolic markers associated with higher egg consumption tended to be associated with lower risk of CVD, MI, and IS (Pearson correlation: −0.64, −0.67, and −0.67, respectively; *Figure 2* and *Supplementary file 1*).

In sensitivity analyses, associations between egg consumption and metabolic markers remained essentially unchanged after further adjustment for hypertension, diabetes, and family history of diabetes or CVD (*Figure 3*). Similar associations were observed when restricting the analyses to participants without any metabolic markers below the limit of detection or rejected by quality control (n=4,251) and using the weekly amount of egg consumption as a continuous independent variable

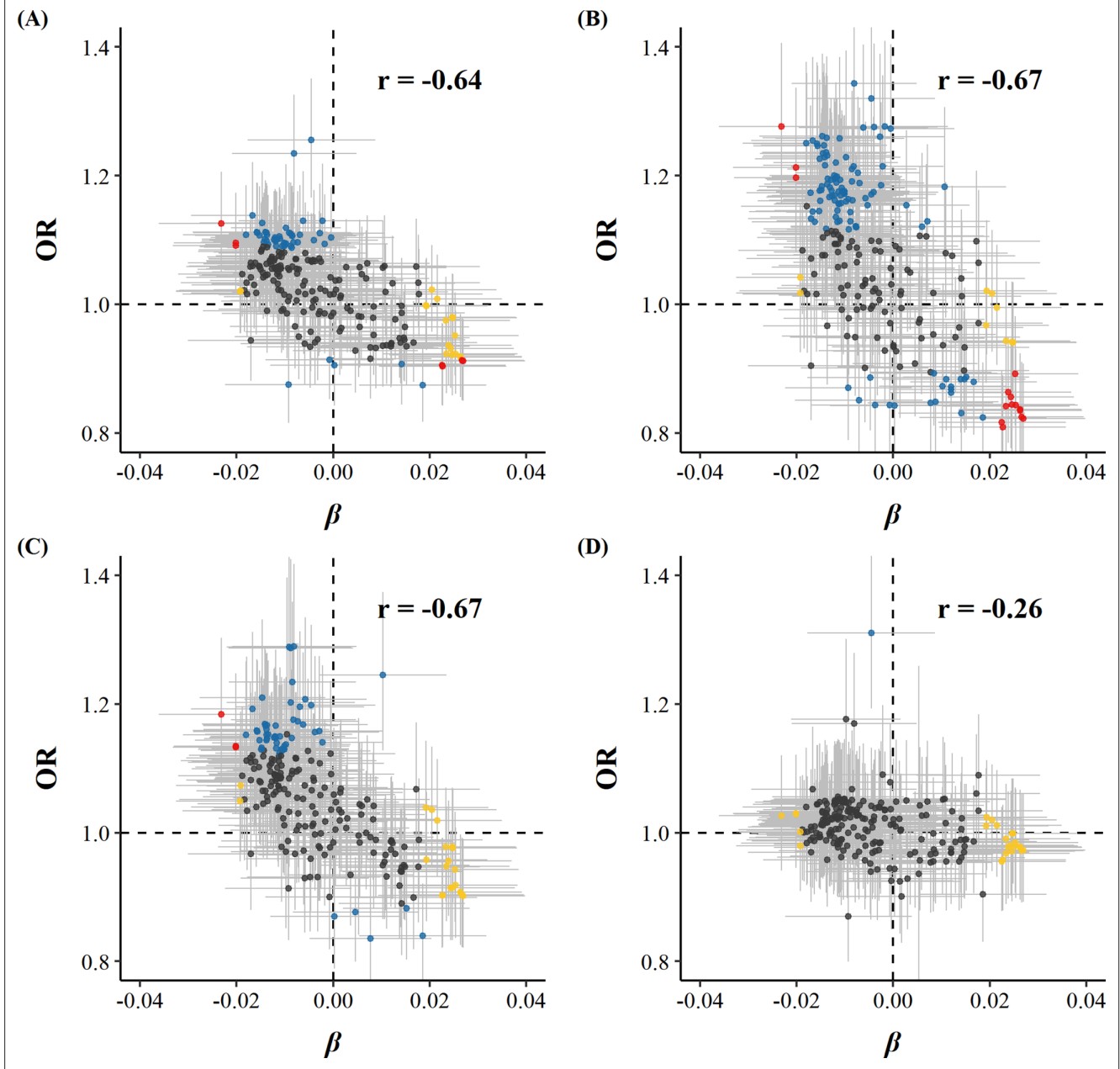

**Figure 2.** Global comparison of SD differences of 225 log-transformed metabolic markers associated with weekly days of egg consumption vs.ORs for (**A**) CVD, (**B**) MI, (**C**) IS, and (**D**) ICH associated with SD higher log-transformed metabolic markers. Yellow dots represented markers that were associated with egg consumption but not with the risk of diseases. Blue dots represented markers associated with the risk of diseases but not with egg consumption. Red dots represented markers associated with both egg consumption and risk of diseases, with overlapping dots darker in color. The gray horizontal and vertical lines represented 95% CI of coefficients and ORs, respectively. Pearson correlations of coefficients and ORs were annotated in the upper right corner. The source data can be found in *Figure 2—source data 1*.

The online version of this article includes the following source data for figure 2:

**Source data 1.** Global comparison of SD differences of 225 log-transformed metabolic markers associated with weekly days of egg consumption and CVD risks.

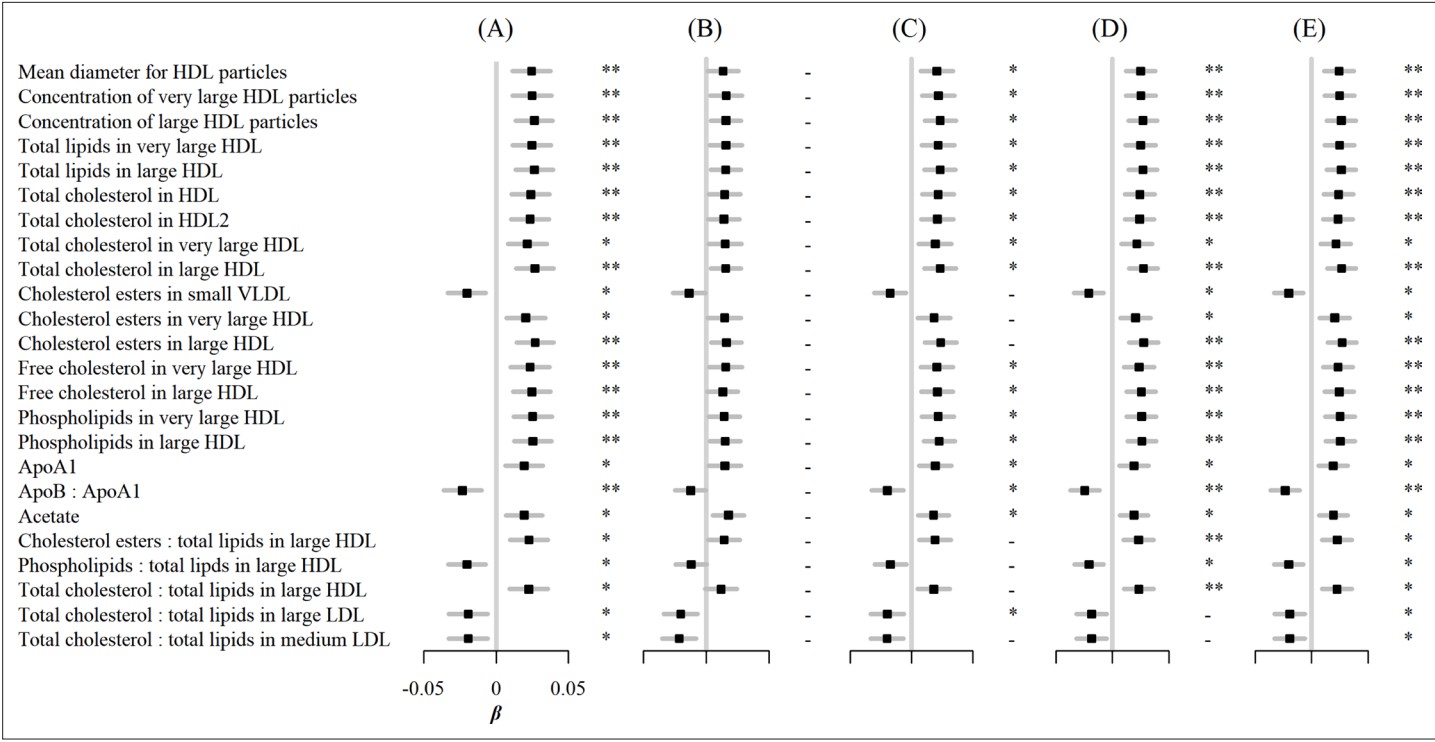

**Figure 3.** Associations of egg consumption and log-transformed metabolic markers. (**A**) In basic models, additionally adjusted for (**B**) BMI, (**C**) hypertension, (**D**) diabetes, and (**E**) family history of diabetes, heart attack or stroke. Significance: * p<0.05, ** p<0.01 (FDR-adjusted p using the Benjamini-Hochberg method).

instead of its frequency (*Figure 4*). However, when BMI was added to the basic model, the associations of egg consumption with metabolites were attenuated toward the null (*Figure 3*).

## Discussion

Based on data of FFQ and NMR metabolomics collected from the CKB population, we simultaneously assessed the associations between self-reported egg consumption and metabolic markers, and further explored whether these markers were associated with the risk of CVD and its subtypes. The results showed that egg consumption was positively associated with apolipoprotein A1, acetate, mean particle diameter of HDL, as well as lipid profiles in very large and large HDL including lipoprotein particle diameters, total lipids, total cholesterol, cholesterol esters, free cholesterol, and phospholipids. In contrast, inverse associations were observed in cholesterol esters in small VLDL, and the ratio of apolipoprotein B to apolipoprotein A1. Moreover, most of these metabolic markers were found to be inversely associated with the risk of CVD. On the whole, 225 metabolic markers showed a pattern. That is, their association with egg consumption and CVD risk were directionally opposite. Our results explained the protective effect of egg consumption on CVD risk through metabolism in the Chinese population.

Results from previous observational studies describing the association between egg consumption and CVD risk are inconsistent. However, few studies have focused on plasma lipid metabolites between such associations so far. Some studies examined the metabonomics characteristics of plasma or feces in a wide range of food groups, including eggs, but had not shown much interest in lipid metabolites (*Guertin et al., 2014*; *Mitry et al., 2019*). Other studies combined eggs with other foods, such as ham, as a feed intervention to investigate consumers' metabolomics characteristics, but the results can not reflect the metabolite changes caused by egg consumption alone (*Rådjursöga et al., 2019*; *Rådjursöga et al., 2017*). This also gives expression to the necessity and urgency of the present study.

Even so, a number of studies and meta-analyses have explored the conventional lipids associated with egg consumption using blood biochemical analyses. Recently, a cohort study based on 39,021

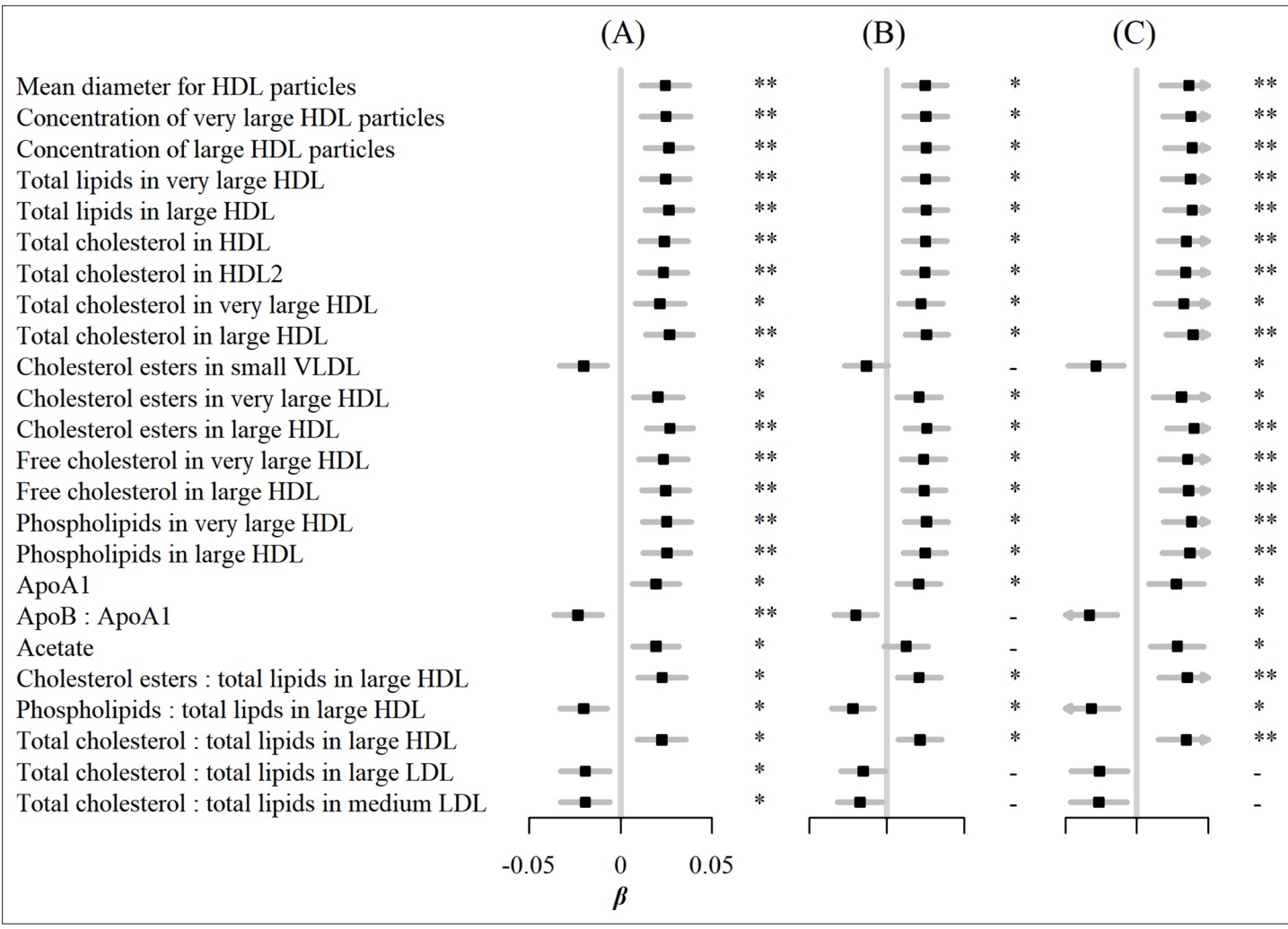

**Figure 4.** Associations of egg consumption and log-transformed metabolic markers. (**A**) In basic models, when (**B**) restricting the analyses to participants without any metabolic markers below the detection limit or rejected by the quality control (n=4251), and (**C**) using weekly amount of egg consumption as continuous independent variable instead of its frequency. Significance: * p<0.05, ** p<0.01 (FDR-adjusted p using the Benjamini-Hochberg method).

overnight fasting adults in China collected egg consumption data and blood lipids biochemical analysis (*Liu et al., 2020*). It was found that compared with the low egg consumption group (<26.79 g/d), medium or high consumption tertiles (>26.79 g/d) had higher levels of HDL-C, which was in line with the present study. It is worth noting that the mediating analysis of that study found that BMI or waist circumference (WC) could partly explain the association between egg consumption and HDL-C. The present study also supported the hypothesis according to sensitivity analysis, in which the associations were no longer significant by additional adjustment of BMI.

Consistent with our results, three small intervention studies based on Americans or Japanese observed positive effects for HDL-C and apolipoprotein A1 (*Kishimoto et al., 2016*; *Missimer et al., 2017*; *Lemos et al., 2018*). In addition, a meta-analysis of 66 eligible randomized clinical trials (RCT) involving 3,185 participants found a non-linear effect for VLDL-C, showing inverse association at the egg consumption level corresponding to the present study (<1.5 eggs/day) (*Khalighi Sikaroudi et al., 2020*).

The debate over egg consumption centered on its effect on cholesterol metabolism. In our study, there was no significant association between egg consumption and lipoprotein in other sizes or densities except small VLDL, very large, and large HDL. To some extent, this reflected the balance between dietary cholesterol absorption and endogenous cholesterol biosynthesis, which accounts for about 25% and 75%, respectively, varying with different food consumption habits

(*Blesso and Fernandez, 2018*; *Lütjohann et al., 2018*; *Kim and Campbell, 2018*). HDL is widely believed to be cardiovascular-friendly lipoproteins responsible for binding and esterifying cholesterol in other tissues for subsequent excretion. The present study suggests that moderate egg consumption increases large HDL levels without altering triglyceride-rich lipoproteins. This mechanism may improve cholesterol metabolic characteristics, thus observing the protective effect on cardiovascular.

Our study provides substantial evidence for the recommendation of appropriate egg consumption in current dietary guidelines in China. The Dietary Guidelines for Chinese Residents recommended that each standard adult should eat 40–50 g eggs a day without discarding the yolk (*Wang et al., 2016*). However, according to the China Statistical Yearbook, the average egg purchase by Chinese residents in 2019 was 10.7 kilograms, or about 29.3 grams per day, which can be a partial estimate of actual consumption. *Statistics NBo, 2020* Although this number has increased over the past decade, it still falls short of dietary guidelines. Hence, more health education and health promotion strategies and policies to encourage moderate egg consumption need to be developed to improve lipid metabolite characteristics in the Chinese population, contributing to CVD prevention.

This study based on the CKB population has many strengths, including relatively large sample size, accurately identified CVD and its subtype events, collection of as many covariates as possible, and the quantification of a wide range of metabolites based on NMR platform, such as lipoproteins and their constituents with varies of size, density, and chemical structure. Besides, the present study restricted the analysis to those free of CVD history and lipid-lowering therapy to minimize the possible confounding or even causal inversion of hypercholesterolemic participants. Our study also had several limitations. First, similar to other large-scale nutritional epidemiological studies, there was an unavoidable recall bias in the FFQ used for estimating egg and other food consumption in our study. There was also measurement bias when five options were used to estimate the weekly days of egg consumption. However, the results of our study were similar to those of previous observational studies or intervention studies, and these results remained robust when the imputed weekly amount of egg consumption was used as the independent variable. Second, even if potential confounding factors such as BMI, frequency of other food consumption, history of chronic diseases, and family history were adjusted in multivariate models or sensitivity analysis, residual confounding due to uncollected or suboptimally collected factors still existed. Third, as the weekly egg consumption of the present study (less than 2 eggs per day) was relatively lower compared with other studies, it is necessary to be cautious when concluding beyond this range. Finally, given the cross-sectional nature, the associations between baseline egg consumption and baseline level of plasma metabolites did not strongly elucidate their causality. Further longitudinal studies are needed to verify the causal roles of lipid metabolites in the association between egg consumption and CVD risk.

## Conclusions

This study set in the Chinese population found significant associations between egg consumption and acetate, lipid-related metabolites within very large and large HDL and small VLDL, and found that these associations were directionally opposite to associations between these metabolites and risk of CVD. These results we reported not only potentially reveal at the small molecule level that lipid metabolism metabolites may play a role in the beneficial effects of egg consumption on CVD, but also provide Chinese population-based evidence for the formulation of strategies and policies to encourage moderate egg consumption.

## Access to research materials/Data sharing

Details of how to access China Kadoorie Biobank data and details of the data release schedule are available from https://www.ckbiobank.org/site/Data+Access.

## Acknowledgements

The most important acknowledgment is to the participants in the study and the members of the survey teams in each of the 10 regional centers, as well as to the project development and management teams based in Beijing, Oxford, and the 10 regional centers.

## Additional information

### Competing interests

The authors declare that no competing interests exist.

### Funding

| Funder | Grant reference number | Author |
| --- | --- | --- |
| National Natural Science Foundation of China | 81973125 | Canqing Yu |
| The Kadoorie Charitable Foundation in Hong Kong | | Liming Li Zhengming Chen |
| National Key Research and Development Program of China | 2016YFC0900500 | Yu Guo |
| Chinese Ministry of Science and Technology | 2011BAI09B01 | Liming Li |
| National Natural Science Foundation of China | 81941018 | Jun Lv |
| National Natural Science Foundation of China | 91846303 | Liming Li |
| National Natural Science Foundation of China | 91843302 | Jun Lv |
| National Natural Science Foundation of China | 81390540 | Liming Li |
| National Natural Science Foundation of China | 81390541 | Liming Li |
| National Natural Science Foundation of China | 81390544 | Jun Lv |
| National Key Research and Development Program of China | 2016YFC0900501 | Yu Guo |
| National Key Research and Development Program of China | 2016YFC0900504 | Canqing Yu |

The funders had no role in study design, data collection and interpretation, or the decision to submit the work for publication.

### Author contributions

Lang Pan, Visualization, Writing - original draft, Writing - review and editing; Lu Chen, Yuanjie Pang, Ling Yang, Iona Y Millwood, Robin G Walters, Yiping Chen, Weiwei Gong, Writing - review and editing; Jun Lv, Conceptualization, Data curation, Methodology, Writing - review and editing; Yu Guo, Pei Pei, Junshi Chen, Conceptualization, Data curation, Investigation, Methodology, Project administration, Writing - review and editing; Huaidong Du, Zhengming Chen, Conceptualization, Data curation, Methodology, Project administration, Writing - review and editing; Canqing Yu, Data curation, Methodology, Project administration, Writing - review and editing; Liming Li, Conceptualization, Data curation, Formal analysis, Investigation, Methodology, Project administration, Writing - review and editing

### Author ORCIDs

Lang Pan http://orcid.org/0000-0002-5231-367X
Jun Lv http://orcid.org/0000-0001-7916-3870
Yuanjie Pang http://orcid.org/0000-0002-4826-8861
Canqing Yu http://orcid.org/0000-0002-0019-0014

### Ethics

Human subjects: The Ethical Review Committee of the Chinese Center for Disease Control and Prevention (Beijing, China, 005/2004) and the Oxford Tropical Research Ethics Committee, University of Oxford (UK, 025-04) approved the study. Before enrolled the study, each participant signed the informed consent and agreed that the data would be used for scientific research and subsequent publication.

### Decision letter and Author response

Decision letter https://doi.org/10.7554/eLife.72909.sa1
Author response https://doi.org/10.7554/eLife.72909.sa2

## Additional files

### Supplementary files

• Supplementary file 1. Detailed association of egg consumption, metabolic markers, and CVD risk. (a) Associations (95% CI) of egg consumption with all 225 log-transformed metabolic markers or devided traits, and of these markers with CVD risk. (b). ORs (95% CI) for MI, IS, and ICH per SD higher log-transformed metabolic markers.

• Supplementary file 2. Members of the China Kadoorie Biobank collaborative group.

• Transparent reporting form

• Source code 1. Analysis code based on Stata software.

### Data availability

For researchers who are interested to access the original data, the access policy and procedures are available at https://www.ckbiobank.org/site/. In brief, the China Kadoorie Biobank (CKB) is being conducted jointly by the Clinical Trial Service Unit (CTSU), Nuffield Department of Population Health, University of Oxford, and Chinese Academy of Medical Sciences (CAMS) in Beijing. Requesters should be employees of a recognized academic institution, health service organization, or charitable research organization with experience in medical research. Requestors should be able to demonstrate, through their peer-reviewed publications in the area of interest, their ability to carry out the proposed study. After registration, details of the required information are provided on the CKB Data Access System. The CKB Access Team will review and respond to data requests within 6-8 weeks. In order to explain more clearly, all the variables required for the analysis of this study are as follows: 1. Background: Basic [Baseline, 1st resurvey, 2nd resurvey], Demographics [Baseline]; 2. Tea consumption: Basic [Baseline], Details [Baseline]; 3. Alcohol consumption: Basic [Baseline], Details [Baseline]; 4. Smoking: Basic [Baseline], Details [Baseline]; 5. Diet: Staple foods [Baseline, 1st resurvey, 2nd resurvey], Animal products [Baseline, 1st resurvey, 2nd resurvey], Vegetables [Baseline, 1st resurvey, 2nd resurvey], Other foods [Baseline, 1st resurvey, 2nd resurvey], Drinks [2nd resurvey]; 6. Medical history: Self-rated [Baseline], Personal [Baseline], Family [Baseline]; 7. Physical activity: Summary - MET [Baseline]; 8. Mental health: Satisfaction [Baseline]; 9. Physical exam: Height and weight [Baseline], Body composition [Baseline], Blood pressure [Baseline], Blood glucose [Baseline]; 10. Biochemistry data: Blood biomarkers (lab) [Baseline], Blood biomarkers (NMR) [Baseline], Blood biomarkers (NMR) - Chylomicrons and extremely large VLDL [Baseline], Blood biomarkers (NMR) - Very large VLDL [Baseline], Blood biomarkers (NMR) - Large VLDL [Baseline], Blood biomarkers (NMR) - Medium VLDL [Baseline], Blood biomarkers (NMR) - Small VLDL [Baseline], Blood biomarkers (NMR) - Very small VLDL [Baseline], Blood biomarkers (NMR) - IDL [Baseline], Blood biomarkers (NMR) - Large LDL [Baseline], Blood biomarkers (NMR) - Medium LDL [Baseline], Blood biomarkers (NMR) - Small LDL [Baseline], Blood biomarkers (NMR) - Very large HDL [Baseline], Blood biomarkers (NMR) - Large HDL [Baseline], Blood biomarkers (NMR) - Medium HDL [Baseline], Blood biomarkers (NMR) - Small HDL [Baseline], Blood biomarkers (NMR) - LDL [Baseline], Blood biomarkers (NMR) - HDL [Baseline], Blood biomarkers (NMR) - fatty acids [Baseline]. We uploaded Stata code that was used to analyze the data. The numbers used to generate Figures 1 and 2 were actually table S1 and S2, which were also uploaded as Excel files. We also uploaded high-resolution raw images (TIF format) of Figures 1 and 2.

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
