## [Editor Report]

Pan et al. investigated associations of self-reported egg consumption with plasma metabolic markers and these plasma metabolic markers with the risk of cardiovascular diseases. In general, there was some impact on metabolic markers which could protect against CVD. The paper will interest scientists in the field of nutritional epidemiology.

---

## [Decision Letter]

**Decision letter after peer review:**

Thank you for submitting your article "Association of Egg Consumption, Metabolic Markers, and Risk of Cardiovascular Diseases: A Nested Case-control Study" for consideration by *eLife*. Your article has been reviewed by 2 peer reviewers, and the evaluation has been overseen by a Reviewing Editor and Balram Bhargava as the Senior Editor. The reviewers have opted to remain anonymous.

Essential revisions:

1. The current study did not consider the difference between hypercholesterolemia patients and participants without hypercholesterolemia. Diagnosed hypercholesterolemia patients preferred to eat less egg, which is the richest source of dietary cholesterol as the authors mentioned. Hypercholesterolemia patients also had a higher risk of cardiovascular diseases. The potential bias should be considered.

2. Whether all the dietary assessments of participants in this study had reliable quality? The authors do not address the information about the exclusion of any unreliable dietary questionnaires.

3. Based on the results of this study, eggs could be a component in a healthy diet. However, the current evidence of egg consumption based on observational studies was not consistent. It is not appropriate to appeal for more health education and health promotion strategies and policies to encourage egg consumption.

4. How did you select the targeted 225 metabolites? Please provide sufficient information in the Introduction section.

5. The study analyzed the associations between plasma metabolic markers and the risk of overall CVD, and three subtypes. Baseline characteristics of these three subtypes should be displayed correspondingly.

6. Cigarette smoking is a strong confounder and should be adjusted in detail.

7. The quality control procedures and rejection criteria need to be included in the Methods section.

8. You have imputed concentrations under LOD with the lowest available concentration, which may lead to underestimation. More widely used approaches, such as imputations using 0 or lowest/2, should be tried.

9. In table 1, what does the content in parentheses after the percentage of some categorical variables mean, such as age, female and ow/ob, etc.? Please indicate this in the table notes. Besides, some continuous variables, such as BMI, SBP and Physical activity, etc., are not shown the standard errors.

10. The standard errors of some variables, including BMI and SBP, should be added in Table 1.

11. In line 145, detailed information on covariables should be described. For instance, how is raw data converted into categorical variables?

12). The possible mechanism of how egg intake affects metabolites and further protects CVD needs to be discussed.

*Reviewer #1 (Recommendations for the authors):*

The manuscript is well written and the statistical methods are valid and correctly applied. Besides, the results are significant and clinical utility in primary prevention of CVD is meaningful since it is proposed by simultaneously explore the associations of self-reported egg consumption with plasma metabolic markers and these markers with the risk of CVD. However, some improvements need to be addressed by the authors.

---

## [Author Response]

Essential revisions:1. The current study did not consider the difference between hypercholesterolemia patients and participants without hypercholesterolemia. Diagnosed hypercholesterolemia patients preferred to eat less egg, which is the richest source of dietary cholesterol as the authors mentioned. Hypercholesterolemia patients also had a higher risk of cardiovascular diseases. The potential bias should be considered.

The authors genuinely understand the reviewers' concern. Before analysis, the possible confounding or even causal inversion of hypercholesterolemic participants was carefully considered. Therefore, we restricted the study to those with no history of CVD and did not take lipid-lowering therapy (line 116-118). The plasma metabolomics assays showed that the average cholesterol level of the 4,778 participants was 3.52±0.61 mmol/L, whose 99% quantile was 5.33 mmol/L. Even at the cut-off point of 5.72 mmol/l, only 20 participants exceeded it, and 14 of them were not fasting for 8 hours. Therefore, in conclusion, we believe that the present study has avoided the confusion caused by hypercholesterolemia as much as possible. Hopefully, this will allay the reviewers' well-meaning concern.

2. Whether all the dietary assessments of participants in this study had reliable quality? The authors do not address the information about the exclusion of any unreliable dietary questionnaires.

We appreciate and understand the reviewer's concern. The present study was as quality-assured as possible on multiple fronts in assessing the participants' habitual dietary intake, added in the Method section of our revised manuscript (line 122-123, 130-133).

First, trained investigators administered the face-to-face laptop-based FFQ with a picture booklet as a reference. Moreover, the laptop used logical warnings to avoid unreasonable answers. For instance, if a participant reported that he never ate any staple food each week, the laptop would pop up a window to double-check. Besides, a separate validation study was conducted from 2015 to 2016 among 432 CKB participants to evaluate the reproducibility and validity of FFQ.^[1]^ It turned out that the weighted Kappa statistic was 0.77 and 0.65 for reproducibility and relative validity of baseline eggs frequency, respectively.

3. Based on the results of this study, eggs could be a component in a healthy diet. However, the current evidence of egg consumption based on observational studies was not consistent. It is not appropriate to appeal for more health education and health promotion strategies and policies to encourage egg consumption.

We agree with the reviewers. In a context where the cardiovascular health effects of egg consumption are still controversial, it is inappropriate, and not the authors' intention, to simply say that egg consumption should be encouraged.

The weekly egg consumption of the present study (<2/d) was relatively lower in Chinese population compared with other studies. As the authors pointed out in the Discussion section, the Dietary Guidelines for Chinese Residents recommend moderate eggs (40-50g/d without discarding the yolk), given their nutritional and controversial nature. Even so, using the purchasing amount in the China Statistical Yearbook as an approximation, Chinese residents' egg consumption (~29g/d) is far from the recommended level. Therefore, given this situation and based on the results of our study, we mentioned encouraging moderate egg consumption.

According to the reviewers' suggestion, the revised manuscript mainly clarifies two points. First, we emphasized "moderate" egg consumption to the basic recommended level, rather than simply encouraging without limit (line 287, 294, and 324). Second, we called for cautions when drawing conclusions at a higher egg consumption level (line 310-312).

4. How did you select the targeted 225 metabolites? Please provide sufficient information in the Introduction section.

We thank the reviewers for asking this question. As expressed in the Introduction section, controversy over egg consumption centered on its effects on circulating lipid metabolites, which had not been systematically assessed. Therefore, we were more interested in lipid-related biomarkers in the egg-metabolite-CVD chain. We selected a targeted NMR-based metabolomics platform that previous large-scale studies have widely used,^[2]^ covering 14 lipoproteins and their subfractions with different densities and particle sizes, as well as other fatty acids, amino acids, and ketone-body-related metabolites. We expected this platform to provide as much biomarker information as possible for the present study. The above has been added in line 88-91.

5. The study analyzed the associations between plasma metabolic markers and the risk of overall CVD, and three subtypes. Baseline characteristics of these three subtypes should be displayed correspondingly.

We thank the reviewer for this kind reminder. The revised manuscript had displayed the baseline characteristics of these three subtypes and controls separately in Table 1.

6. Cigarette smoking is a strong confounder and should be adjusted in detail.

The authors entirely agree with the reviewer's opinion and consider smoking status one of the critical confounding factors. We want to clarify that according to the frequency, type, and quantity of smoking obtained from the questionnaire, smoking status was converted into a categorical variable (5 categories: never or occasional smoker, former smoker, ≤10 cigarettes, ≤20 cigarettes, or more per day) in the basic model of the present study (added at line 170-171). Through this detailed classification, rather than a binary variable, we hope to avoid as much confusion as possible from smoking status. Similarly, detailed information on other covariates was also added in the revised manuscript.

7. The quality control procedures and rejection criteria need to be included in the Methods section.

We thank the reviewers for the professional advice on metabolomics assays. In order to make NMR metabolomics quality control procedures more transparent, we cited an article that detailedly described these procedures^[3]^ and added brief information in the Method section of the revised manuscript, including liquid handling, quality control samples, sample identification, and so on (line 142-149).

8. You have imputed concentrations under LOD with the lowest available concentration, which may lead to underestimation. More widely used approaches, such as imputations using 0 or lowest/2, should be tried.

We agree with the reviewer that the imputation method of missing values under LOD may affect the study results. The replacement with a nonzero value is widely used, including minimum, mean, or median values between 0 and the detection limit, none of which has been proven perfect. Theoretically, given the high detection rate in the present study (only 0.8% of concentrations in total were under LOD), the imputation method is unlikely to overturn the results. Even so, the authors tried to reanalyze using the half minimum method at the reviewers' suggestion. The results showed that the metabolites associated with egg consumption remained essentially unchanged, although the three lipid ratios were no longer statistically significant after FDR correction (FDR-*p* was 0.11, 0.07, and 0.09, respectively).

9. In table 1, what does the content in parentheses after the percentage of some categorical variables mean, such as age, female and ow/ob, etc.? Please indicate this in the table notes. Besides, some continuous variables, such as BMI, SBP and Physical activity, etc., are not shown the standard errors.

Thank the reviewers for pointing it out. The authors apologize for our carelessness. We rechecked Table 1, which showed continuous variables as the means (with SEs in parentheses) while categorical variables as percentages.

10. The standard errors of some variables, including BMI and SBP, should be added in Table 1.

We thank the reviewers for this reminder. In revised Table 1, all continuous variables' standard errors (SE) had been added.

11. In line 145, detailed information on covariables should be described. For instance, how is raw data converted into categorical variables?

Based on suggestions from reviewers, detailed information on covariables was added in the Method section of our revised manuscript (line 162-174).

12). The possible mechanism of how egg intake affects metabolites and further protects CVD needs to be discussed.

The authors appreciate and fully agree with the reviewers' suggestion. Based on the body's balance of endogenous and exogenous metabolites, as well as the biological function of HDL, the hypothesis that moderate egg consumption improves circulating cholesterol metabolism was added in the Discussion section (line 277-286), given the observed rising levels of large HDL but not triglyceride-rich lipoprotein in the present study.

References

1. Qin C, Guo Y, Pei P*, et al.* The Relative Validity and Reproducibility of Food Frequency Questionnaires in the China Kadoorie Biobank Study. Nutrients. 2022;14(4):794.

2. Wurtz P, Kangas AJ, Soininen P*, et al.* Quantitative Serum Nuclear Magnetic Resonance Metabolomics in Large-Scale Epidemiology: A Primer on -Omic Technologies. Am J Epidemiol. 2017;186(9):1084-96. DOI:10.1093/aje/kwx016.

3. Soininen P, Kangas AJ, Wurtz P*, et al.* Quantitative serum nuclear magnetic resonance metabolomics in cardiovascular epidemiology and genetics. Circ Cardiovasc Genet. 2015;8(1):192-206. DOI:10.1161/CIRCGENETICS.114.000216.